# Publication speed in pharmacy practice journals: A comparative analysis

**Antonio M. Mendes**[1], **Fernanda S. Tonin**[1], **Felipe F. Mainka**[1], **Roberto Pontarolo**[2], **Fernando Fernandez-Llimos**[3,4]*

**1** Pharmaceutical Sciences Postgraduate Program, Federal University of Paraná, Curitiba, Brazil, **2** Department of Pharmacy, Federal University of Paraná, Curitiba, Brazil, **3** Faculty of Pharmacy, Laboratory of Pharmacology, University of Porto, Porto, Portugal, **4** Center for Health Technology and Services Research (CINTESIS), University of Porto, Porto, Portugal

* fllimos@ff.up.pt

**Data Availability Statement:** Data available at https://doi.org/10.17605/OSF.IO/QY2AE.

**Funding:** The authors received no specific funding for this work.

## Abstract

### Background

Scholarly publishing system relies on external peer review. However, the duration of publication process is a major concern for authors and funding bodies.

### Objective

To evaluate the duration of the publication process in pharmacy practice journals compared with other biomedical journals indexed in PubMed.

### Methods

All the articles published from 2009 to 2018 by the 33 pharmacy practice journals identified in Mendes et al. study and indexed in PubMed were gathered as study group. A comparison group was created through a random selection of 3000 PubMed PMIDs for each year of study period. Articles with publication dates outside the study period were excluded. Metadata of both groups of articles were imported from PubMed. The duration of editorial process was calculated with three periods: acceptance lag (days between 'submission date' and 'acceptance date'), lead lag (days between 'acceptance date' and 'online publication date'), and indexing lag (days between 'online publication date' and 'Entry date'). Null hypothesis significance tests and effect size measures were used to compare these periods between both groups.

### Results

The 33 pharmacy practice journals published 26,256 articles between 2009 and 2018. Comparison group random selection process resulted in a pool of 23,803 articles published in 5,622 different journals. Acceptance lag was 105 days (IQR 57–173) for pharmacy practice journals and 97 days (IQR 56–155) for the comparison group with a null effect difference (Cohen's d 0.081). Lead lag was 13 (IQR 6–35) and 23 days (IQR 9–45) for pharmacy practice and comparison journals, respectively, which resulted in a small effect. Indexing lag was 5 days (IQR 2–46) and 4 days (IQR 2–12) for pharmacy practice and control journals, which

**Competing interests:** FFL and FST are editors of the journal Pharmacy Practice, but this does not alter our adherence to PLOS ONE policies on sharing data and materials. The other authors have no conflict of interest.

also resulted in a small effect. Slight positive time trend was found in pharmacy practice acceptance lag, while slight negative trends were found for lead and indexing lags for both groups.

## Conclusions

Publication process duration of pharmacy practice journals is similar to a general random sample of articles from all disciplines.

## Introduction

Publishing articles is the most common way to disseminate the results of research activities. The old "publish or perish" culture is fully alive [1]. Publication times have always been a major concern for authors because a delay in the publication leads to a delay in the dissemination of the knowledge they discovered and because their careers and future funding depend on publications [2]. These delays are amplified because many articles are peer reviewed and rejected at least once before they are finally published [3].

Authors pressure editors to have their manuscripts published quickly, and funding bodies analyze the times between their investment (funding) and their profit (publication). The Australian National Health and Medical Research Council reported that among the 77 RCTs funded between 2008 and 2010, 72% had their main results published after 8 years, with a median delay of 7.1 years after funding (95% CI 6.3 to 7.6) [4]. The U.S. National Heart, Lung, and Blood Institute (NHLBI) reported that the results of 42% of the 232 RCTs funded between 2000 and 2011 had been published before January 2013, with a median time from funding to publication of 25 months [5]. In a secondary analysis, the NHLBI identified the amount funded and the use of intention-to-treat analyses as the main variables that influence the speed at which the results of the trials are published [6]. In a recent analysis of 1509 trials that were registered in a clinical trials registration database with a completion time between 2009 and 2013 and that had a German university medical center as a leading institution, only 39% of the trials had their results published "in a timely manner (<24 months after completion)", with a median publication delay time between 28 and 36 months [7].

The editorial process is a compilation of sequential steps with the final aim of ensuring the quality of the papers published. Fig 1 presents the different steps in the scholarly publication process and provides the names for the delays or lags following Lee et al.'s terminology [8]. Immediately after receiving a manuscript, editors face the responsibility of using the controversial measure of a "quick and brutal" desk rejection of potentially weak or irrelevant manuscripts [9]. If the submission is not desk rejected, the external peer review system starts and frequently involves with a two-round process. Peer review, which has been described as "crude and understudied, but indispensable", is an essential part of the evaluation of publishability [10]. However, authors usually dislike the results of this process [11]. Although the duration of the editorial process is frequently criticized, better alternatives have not been created [12]. Before establishing peer review as a requirement for all their manuscripts, even the journals with the strongest reputations were accused of biased selection [13]. During emergency times, some journals have reported reducing the requirements by asking "peer reviewers to be especially vigilant in guarding against unreasonable demands for revision that are not essential to the main conclusions of a manuscript" [14,15].

Trying to reduce publication times is not a new initiative. In fact, approximately 100 years ago, well-known scientists started using Letters to the Editor as a way to reduce the time to get

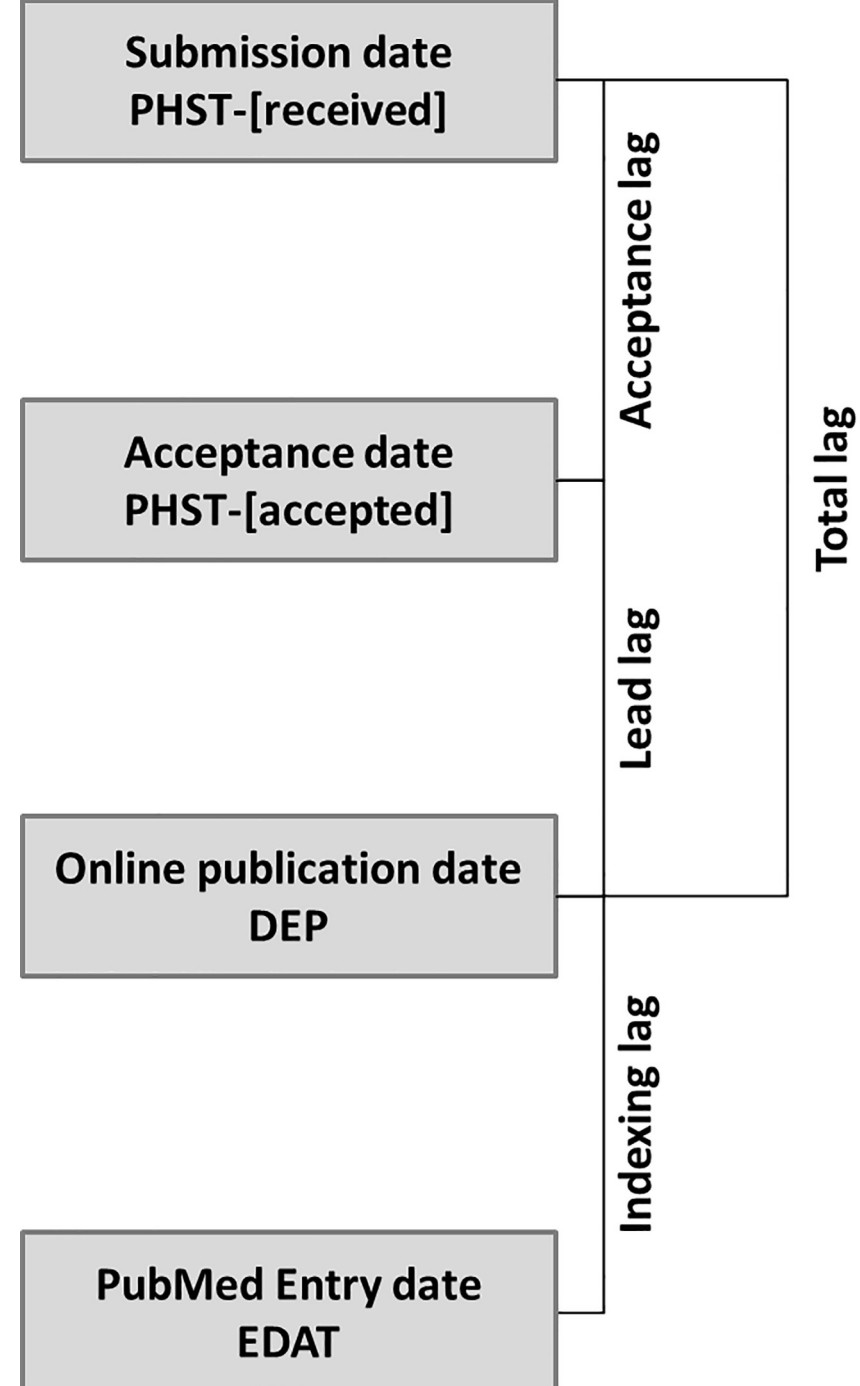

**Fig 1. Editorial process description with terms used in this study.**

their ideas published [16]. The advent of electronic publication seemed to solve the problem of publication delays but was soon revealed to be an incomplete solution [17]. Improving publication speed has become a major aim of journal editors [18]. However, a balance is needed between the urgency for obtaining the publication goal and the reliability of the scholarly publishing system based on rigorous peer review, as recently demonstrated [19].Pharmacy practice

is a scientific discipline that suffers from a similar situation as other scientific areas and has a similar aim to have manuscripts published quickly. Rodriguez analyzed the time-to-index of three pharmacy practice journals between 2010 and 2011 and found a median time of 114 days (IQR, 98–141 days) [20]. Irwin and Rackham compared the time-to-index of pharmacy journals (158 days; SD 58) with the time-to-index of medicine journals (185; SD 96) and nursing journals (234; SD 107) [21]. In a new analysis of medical, nursing, and pharmacy journals entered in PubMed in 2012 and 2013, Rodriguez reported different results, with median times-to-index of 62 days (IQR 45–89), 187 days (IQR 148–255), and 280 days (IQR 134–400) for medical, pharmacy, and nursing journals, respectively [22]. However, to the best of our knowledge, no widescale comparison of publication times between pharmacy practice journals and other biomedical disciplines has been performed.

The aim of the study was to evaluate the duration of the publication process in pharmacy practice journals compared with the publication process duration in a random sample of articles indexed in PubMed.

## Materials and methods

### Data collection

A list of pharmacy practice journals was obtained from Mendes et al.'s study [23]. That study objectively classified the 285 pharmacy journals into six clusters, namely, 'Cell Pharmacology' (20 journals), 'Molecular Pharmacology' (46 journals), 'Clinical Pharmacology' (57 journals), 'Pharmacy Practice' (67 journals), 'Pharmaceutics' (35 journals) and 'Pharmaceutical Analysis' (60 journals). Of the 67 pharmacy practice journals, 33 are indexed in the U.S. National Library of Medicine (NLM) PubMed (https://www.pubmed.org). On February 2019, metadata of all the articles published in these 33 journals between 2009 and 2018 (ten years) were extracted from PubMed to create a pool of 'pharmacy practice journal' articles. Articles published in each year for each journal were retrieved from PubMed using the International Standard Serial Number (ISSN) of each journal with the [IS] field descriptor combined with the 'AND' Boolean operator to the year using the [DP] field descriptor.

To create a biomedical journals comparison group of randomly selected articles published in non-pharmacy practice journals, the first PMID [name used by the National Library of Medicine for the PubMed Unique Identifier] of each year between 2009 and 2018 were identified in PubMed. PubMed was selected because is the biggest bibliographic database of biomedical literature. The required sample size was calculated after a preliminary analysis of 12,380 randomly selected articles extracted from PubMed, which resulted in a mean delay of 125 days (SD 98) between article reception and publication, with 43.4% of the articles providing data for this calculation. Aiming to identify a 10-day (less than 10% the delay of the whole editorial process in the preliminary analysis) between-group mean difference in each year with an alpha error of 0.05 and a power of 80%, a sample of 1509 articles was obtained, and this sample size was calculated using G*Power (University of Kiel, Kiel). Considering the 50% of potentially incomplete metadata or inexistent PMIDs, a sample of 3000 PMIDs per year was created using the Research Randomizer website (https://www.randomizer.org). To obtain each set of randomly generated numbers, first and last PMID of each year were used as 'number range' limits, setting the 'remain unique' and the 'markers-off' options of the randomizer.org webpage. Data from the comparison group selection process are available in Supporting information–S1 Table in S1 Appendix. In February 2019, the metadata of the articles indexed with the randomly generated PMIDs were extracted from PubMed to create a pool of comparison biomedical journal articles using an identical metadata extraction process to the used for the pharmacy practice journals.

Articles were excluded from the analyses if the entry date [date when the metadata were introduced in PubMed] was between the study limits but the actual date of publication was outside the study period.

## Data process

PubMed records of all the articles in both groups were imported into an EndNote X4 (Thomson Reuters, Toronto) and then exported into an Excel spread sheet (Microsoft, Redmond). Submission date (when the journal received the manuscript) was obtained from the PubMed field PHST-[received]; acceptance date (when the journal communicated the acceptance to the authors) was obtained from PHST-[accepted]; online publication date (when the journal made the article available in journal's webpage) was obtained from the field DEP; entry date (when PubMed processes the metadata submitted by the publisher and made them available for searches) was obtained from the field EDAT; publication language was obtained from the field LA; and publication country was obtained from the field PL.

The CiteScore for 2018, percentile of the journal in 2018 CiteScore distribution and Scopus Sub-Subject Area were obtained from the Scopus Sources database (https://www.scopus.com/sources). Each journal's Impact Factor (IF) for 2017 was obtained from the Journal Citation Reports, available through the Web of Science (https://jcr.clarivate.com). Comparison group journal business model was classified as 'open access' if the publishing journal was included in the Directory of Open Access Journals (DOAJ) list (https://doaj.org/csv) obtained in October 2019. For pharmacy practice journals, the condition of open access was assigned by inspection of journals' websites.

Four lag time measures for the different steps of the publication and indexing process were calculated: total publication lag (days between 'submission date' and 'online publication date'), acceptance lag (days between 'submission date' and 'acceptance date'), lead lag (days between 'acceptance date' and 'online publication date'), and indexing lag (days between 'online publication date' and 'entry date'). Fig 1 depicts these time intervals. We have not considered an additional exiting lag time, the cataloging lag (time between the 'entry date' and the Medical Subject Heading (MeSH) allocation date [MHDA] because it would only be applicable to journals indexed in MEDLINE, not only in PubMed.

## Data analysis

Categorical variables were presented as absolute values and frequencies. For continuous variables, normality was assessed through the Kolmogorov-Smirnov test with additional visual inspection of the Q-Q plot. Because of the poor metadata reporting rate identified in the preliminary analyses, we preferred not using any method to impute missing values due to the potential low reliability. Associations between two categorical variables were tested with the chi-squared test and reported with odds ratio and 95% confidence intervals. Correlations between two non-normal variables were calculated with Spearman's rho. Comparisons between two independent non-normally distributed variables were conducted with the Mann-Whitney test. To calculate effects size measures of the differences between the two groups, which are preferred to null hypothesis test results, the U-statistic was converted into Cohen's d in accordance with recommendations from Cohen [24] and Fritz et al. [25] and using the Psychometrica calculator (https://www.psychometrica.de/effect_size.html). Effect sizes were categorized as follows: <0.1 no effect, 0.1–0.4 small effect, 0.5–0.7 intermediate effect, and > 0.7 large effect [24]. Data were analyzed using SPSS v20 (IBM, Armonk) and RStudio v1.2 (RStudio Inc., Boston). Linear multivariate regressions were also performed for the three time lags, using group, IF, CiteScore, and open access condition as independent variables. Variance inflation factor (VIF) was used to estimate collinearity in multivariate models.

**Table 1. Articles selected for the study.**

| year | Publication dates | | |
|---|---|---|---|
| | Non-pharmacy practice | Pharmacy practice | Total |
| 2009 | 1,554 | 1,637 | 3,191 |
| 2010 | 1,677 | 2,018 | 3,695 |
| 2011 | 1,866 | 2,103 | 3,969 |
| 2012 | 3,567 | 2,462 | 6,029 |
| 2013 | 2,628 | 2,560 | 5,188 |
| 2014 | 1,311 | 2,743 | 4,054 |
| 2015 | 1,376 | 3,024 | 4,400 |
| 2016 | 2,071 | 3,037 | 5,108 |
| 2017 | 3,556 | 3,312 | 6,868 |
| 2018 | 4,197 | 3,360 | 7,557 |
| | 23,803 | 26,256 | 50,059 |

## Results

The 33 pharmacy practice journals indexed in PubMed published a total of 26,256 articles between 2009 and 2018. These articles were published in three languages, English (90.0%), Japanese (8.1%), and French (1.9%). The distribution of these articles per journal and year is available in Supporting information–S2 Table in S2 Appendix. The CiteScore was calculated by Scopus for 25 of the 33 pharmacy journals indexed in PubMed; the median CiteScore was 1.09 (interquartile range, IQR 0.80–1.73). Only 8 journals had an IF; the median IF was 2.094 (IQR 1.505–2.688).

The 3,000 PMIDs randomly selected per year for the study period resulted in a total of 25,272 existing PMIDs, which led to 23,803 articles for the comparison group after excluding those published outside of the study period (Table 1). These articles were published in 5,601 different journals with a median of 2 articles published per journal (IQR 1–5); PLoS ONE was the most prevalent journal, with 471 articles. These articles were published in 27 different languages, with English (22,565 articles; 94.8%) and Chinese (335 articles; 1.4%) being the most common languages. The journals publishing these articles were published in 75 countries, with the United States (9,775 articles), the United Kingdom (5,883 articles), and the Netherlands (1,604 articles) as the most common countries. The CiteScore was calculated for 4,856 of the 5,601 journals, and the median CiteScore was 2.06 (IQR 1.20–3.20). A total of 3,835 journals had an IF; the median IF was 2.396 (IQR 1.538–3.607).

Reporting rates of the publication process dates were significantly lower in the pharmacy practice articles than in the comparison group, with odds ratios between 0.30 and 0.53 (Table 2). This lower rate of reporting the publication process dates was mainly associated with the pharmacy practice journals published in the U.S., U.K. and Japan; pharmacy practice

**Table 2. Frequencies of article processing dates reported.**

| | Pharmacy practice | Non-pharmacy practice | p-value[*] | Odds ratio (95%CI) |
|---|---|---|---|---|
| Submission date | 8,900 (33.9%) | 11,656 (49.0%) | <0.001 | 0.53 (0.52–0.56) |
| Acceptance date | 8,964 (34.1%) | 12,106 (50.8%) | <0.001 | 0.50 (0.48–0.52) |
| Online publication date | 9,469 (36.1%) | 15,557 (65.4%) | <0.001 | 0.30 (0.29–0.31) |

[*]chi-square test.

(Data presented as number and percentage).

**Table 3. Lag times (days) calculated in the two groups.**

| | Pharmacy practice | | | Non-pharmacy practice | | | p-value** | Cohen's d |
|---|---|---|---|---|---|---|---|---|
| | n | Median | IQR | n | Median | IQR | | |
| Acceptance lag* | 8,884 | 105 | 57–173 | 11,146 | 97 | 56–155 | <0.001 | 0.081 |
| Lead lag* | 7,100 | 13 | 6–35 | 10,559 | 23 | 9–45 | <0.001 | 0.335 |
| Total lag* | 7,086 | 138 | 79–217 | 9,899 | 131 | 82–197 | 0.002 | 0.048 |
| Indexing lag* | 9,189 | 5 | 2–46 | 14,266 | 4 | 2–12 | <0.001 | 0.496 |

*Kolmogorov-Smirnov<0.001

**Mann-Whitney test; IQR = Inter-quartile range.

journals published in France, the Netherlands, Saudi Arabia, Spain and Switzerland have higher reporting rates than journals from these countries from the comparison group (Supporting information –S3 Tables 1 to 4 in S3 Appendix). Among the pharmacy practice articles, 7,086 (27.0%) reported all three dates, while 9,917 of the comparison articles (41.7%) reported all four dates (OR 0.52; 95% CI 0.50–0.54).

Due to the large sample size used, a significant difference was found in the lag times analyzed between the pharmacy practice group and the comparison group. However, the effect sizes of these differences ranged from no effect (Cohen's d = 0.05) to small effect (Cohen's d = 0.40) (Table 3). The acceptance lag slightly increased over time in pharmacy practice journals (Spearman's rho = 0.163; p<0.001), while the trend was nearly flat in the comparison group (rho 0.045; p<0.001) (Fig 2). The lead lag decreased over time in both groups, rho = -0.230 in pharmacy practice journals and rho = -0.127 in comparison journals (Fig 3). The indexing lag remained stable over time in both groups (Fig 4). Detailed data are provided in Supporting information–S4 Tables 1 to 4 in S4 Appendix.

When only pharmacy practice journals were analyzed, important variations in the acceptance lag were found, ranging from medians of 46 days (IQR 27–80) in J Basic Clin Pharm, 48 days (IQR 32–71) in Pharmacy (Basel), 290 days (IQR 230–349) in Curr Pharm Teach Learn and 242 days (IQR 182–304) in Int J Pharm Pract. A trend analysis of the acceptance lag demonstrated a steady trend in the majority of the pharmacy practice journals during the study period; a few journals presented trends with significant correlations, but with no effect or small effect sizes. Only one journal, namely, Pharmacy (Basel), presented a moderate effect with a decrease in acceptance lag over time (rho = -0.436); this journal had the lowest acceptance lag among all the pharmacy practice journals in 2018, at 40 days (IQR 25–55). The complete data for processing lag times per year with the corresponding violin plots of pharmacy practice journals are available in Supporting information–S 5. The lead lag presented a significant correlation decrease during the study period for almost all the pharmacy practice journals; there was a large effect size (Spearman's rho >0.7) only in Res Social Adm Pharm. The majority of the pharmacy practice journals also presented a significant correlation decrease in the indexing lag over time, but four journals presented high indexing lag times with medians greater than 100 days (Curr Pharm Teach Learn, Hosp Pharm, J Young Pharm, and Saudi Pharm J).

Compared with subscription journals, open access pharmacy practice journals presented a lower acceptance lag (72 days, IQR 46–118 vs. 128 days, IQR 72–203, p<0.001; OR = 0.612), a lower lead lag (10 days, IQR 6–26 vs. 14 days, IQR 7–37, p<0.001: OR = 0.172) and a greater indexing lag (92 days, IQR 14–276 vs. 5 days, IQR 2–25, p<0.001; OR = 0734). These differences in lag times between open access and subscription journals were nonexistent in the comparison group, with acceptance lags of 100 days (IQR 58–160) vs. 98 days (IQR 57–154)

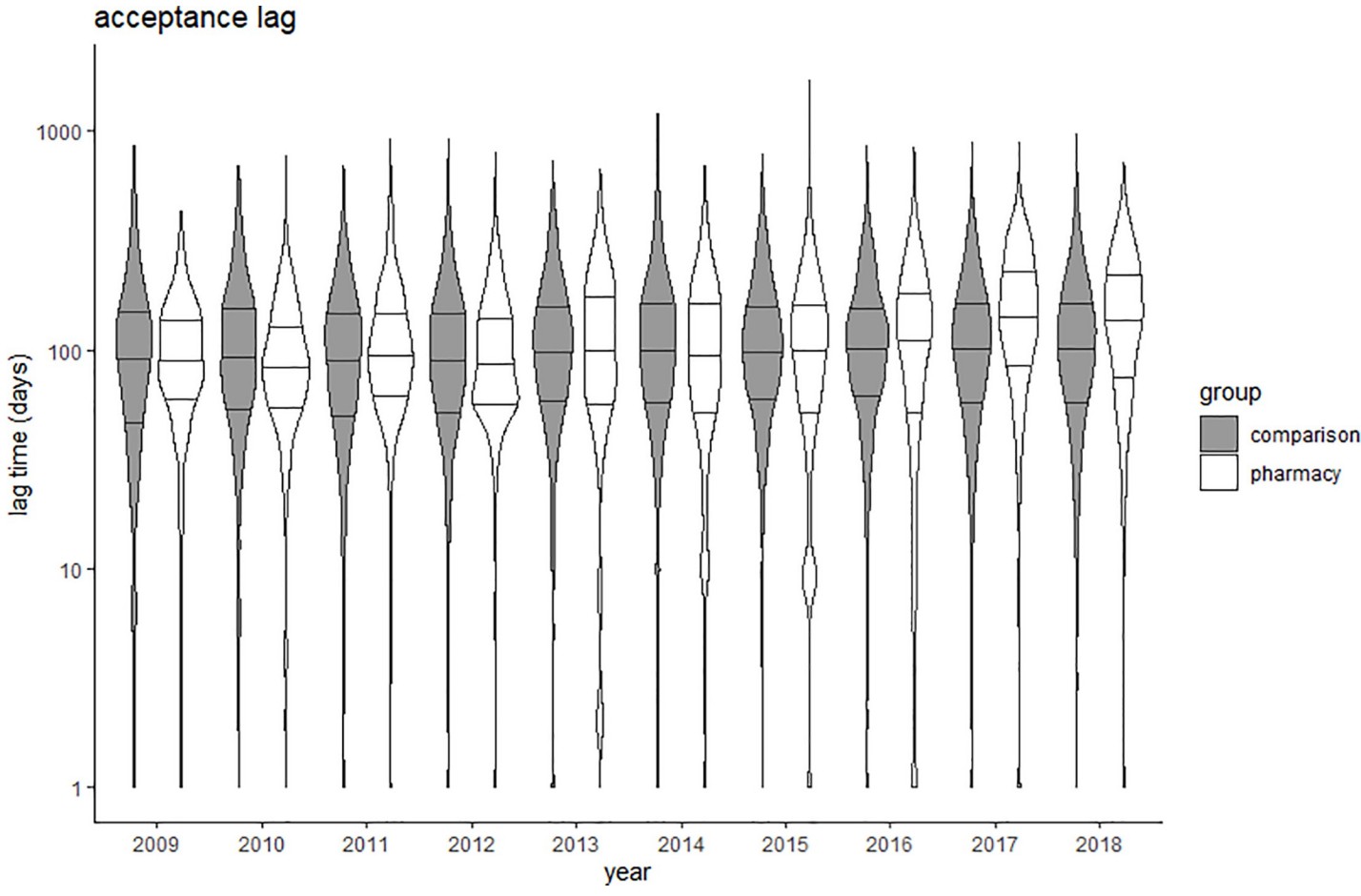

**Fig 2. Violin plots of acceptance lag (time from submission to acceptance) in comparison (grey) and pharmacy practice (white) journals.** Y-axis presented in logarithmic scale; Crossbars represent median and 25% and 75% quartiles.

(p = 0.056; OR = 0.037), lead lags of 23 days (IQR 11–42) vs. 22 days (IQR 9–46) (p = 0.607, OR = 0.010), and indexing lags of 4 days (IQR 2–18) vs. 3 days (IQR 2–6) (p<0.001; OR = 0.169), respectively.

The acceptance lag was slightly associated with the CiteScore in the pharmacy practice group (rho = -0.131; p<0.001) and not the comparison group (p = 0.255). Impact factor presented also a slight inverse association with the acceptance lag in pharmacy practice journals (rho = -0.195, p<0.001) and in the comparison group (rho = -0.082, p<0.001); however, the effect sizes indicated no effects.

Multivariate analyses confirmed the associations found in the bivariate analyses for the three lag times (Table 4). Moderate VIF was found for CiteScore and IF, but always below the commonly accepted cutoff (VIF: 5).

## Discussion

Our 10-year analysis of more than 23,000 articles published in pharmacy practice journals compared with a random sample of more than 20,000 articles published in other journals in the same period of time indicated that there was a very similar acceptance lag (time to accept a manuscript) between the two groups; however, there was a slightly lower lead lag (time to

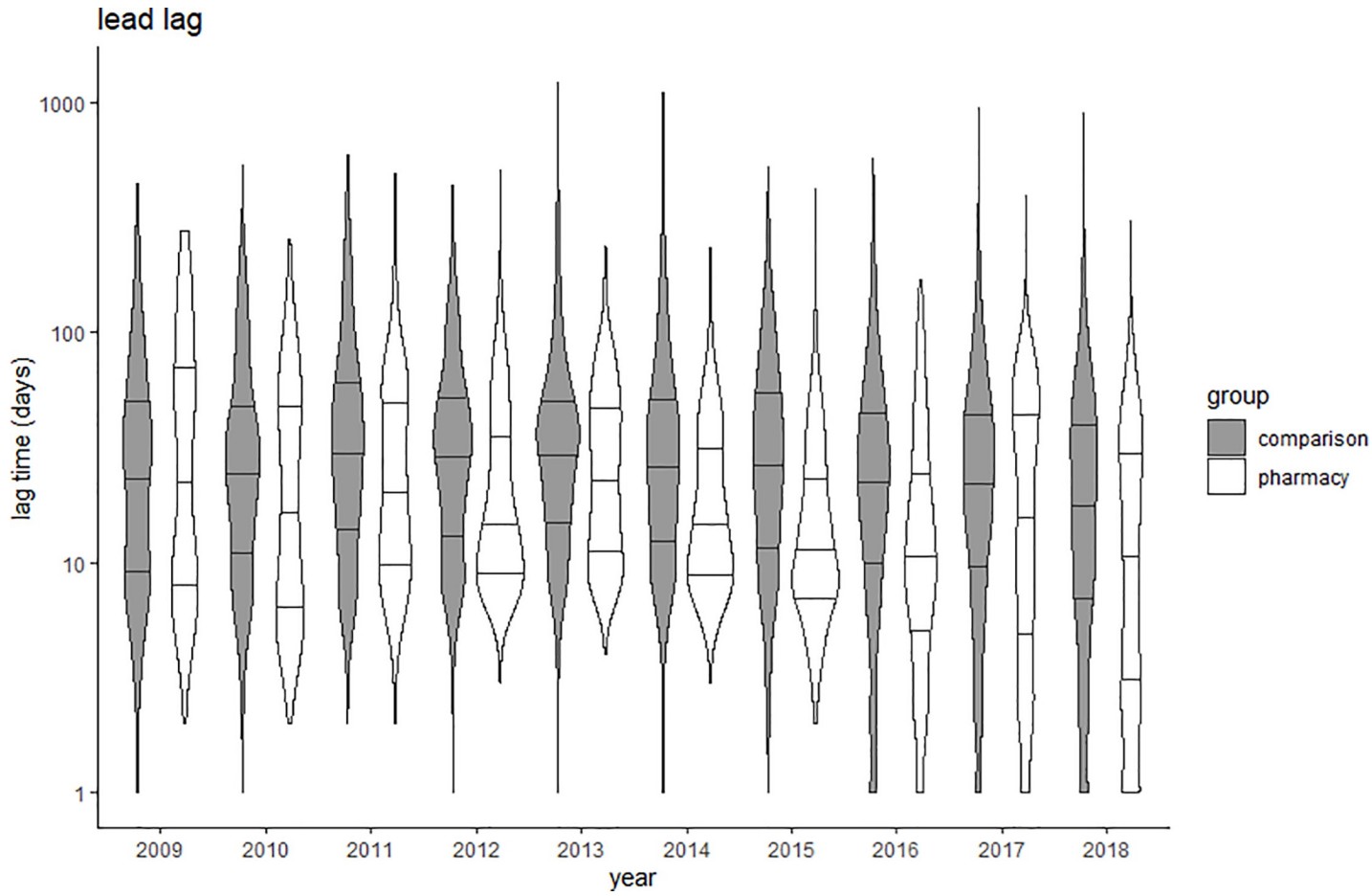

**Fig 3. Violin plots of lead lag (time from acceptance to online publication) in comparison (grey) and pharmacy practice (white) journals.** Y-axis presented in logarithmic scale; Crossbars represent median and 25% and 75% quartiles.

make the article available online) and a slightly longer indexing lag (time to index the article in PubMed) in the pharmacy journals than in the comparison group.

Our results showed that, once a manuscript is accepted by the journal, an article will be published online in approximately 13 days in pharmacy practice journals and 23 days in other journals. An additional five days is necessary to make the article available in PubMed. Both, lead and indexing lags in pharmacy practice journals and in the generic comparison group were smaller than those reported by Lee et al. among the Korean medical journals [8]. It seems that these post-acceptance delays are not important for authors when they ask for a more rapid editorial process, because at this point, they have already obtained a communication from the editor about the article's acceptance [26].

Authors of articles both in pharmacy practice journals and in the comparison journals had to wait approximately 100 days to have their manuscript accepted. This acceptance lag is similar to the one reported by Lee et al. among Korean medical journals of 102 days [8]. About 100 days of acceptance lag was also reported in the Himmelstein analysis in which a quite flat historical trend was demonstrated [27]. Himmelstein also reported an analysis of Public Library of Science journals with an average 100 days acceptance lag among these journals [28]. This time is the main reason that authors complain about the slow editorial process, especially when they include the time spent in the previous editorial processes with negative feedback

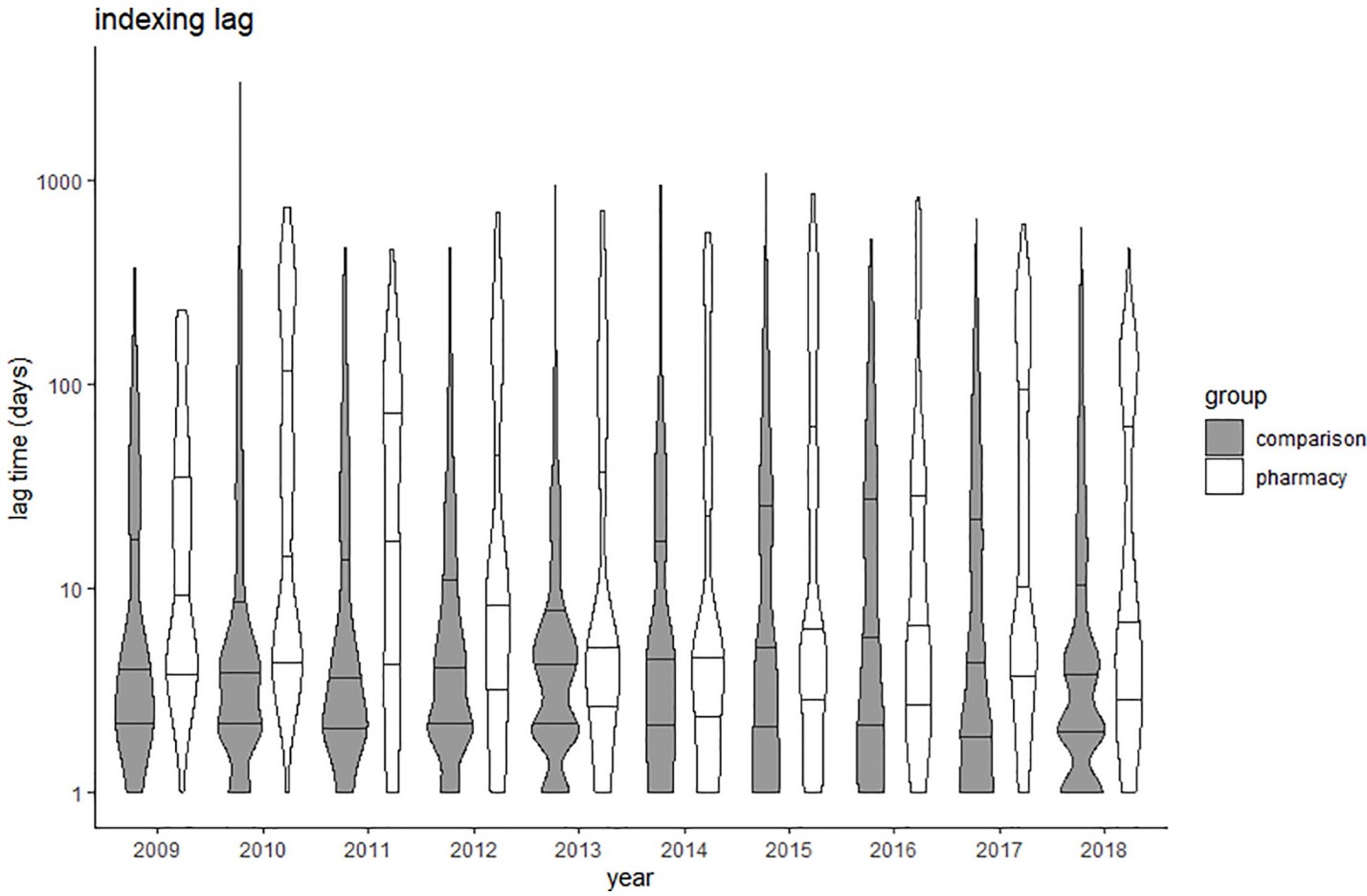

**Fig 4. Violin plots of indexing lag (time from online publication to PubMed indexing) in comparison (grey) and pharmacy practice (white) journals.** Y-axis presented in logarithmic scale; Crossbars represent median and 25% and 75% quartiles.

[26]. It is important to consider that this time period of more than three months includes the time that editors and reviewers take to scrutinize the manuscript, the time authors need to modify the original text to fulfill reviewers' recommendations, and the time editors need to find a sufficient number of peer reviewers for the manuscript. The latter aspect has become a major concern for journal editors, with two to four invited reviewers needed for one acceptance [29,30].

Several solutions have been suggested to reduce the acceptance lag, usually related to reducing or eliminating the time devoted to the peer review stage, but the effectiveness of these solutions has not yet been demonstrated [31]. Stern and O'Shea suggested several strategies to move on from an "outdated publishing process" to an author-based decision process, where authors and not editors decide whether a manuscript should be published or not, and the "curation" process starts after the publication of the article [32]. This is the rationale of the pre-print servers: repositories where authors deposit their manuscripts, which become immediately indexed and available to the general public with no prepublication peer review [33]. Scientific fields such as physics commonly use rapid publication systems to share ideas. However, health sciences and especially clinical sciences should carefully approach these "pre-refereed research outputs" [33]. Clinical decisions made by professionals and self-care decisions made by lay people are frequently made based on published articles, which may pose health

**Table 4. Multivariate analyses for the three lag times.**

| Acceptance lag | R-square: 0.006; Durbin-Watson = 1.788 | | | |
|---|---|---|---|---|
| | Beta | 95%IC | p-value | VIF |
| Study group | -3.910 | -7.077: -0.742 | 0.016 | 1.072 |
| CiteScore (2018) | 1.000 | -0.267: 2.268 | 0.122 | 2.991 |
| Opean Access journal? | -13.822 | -17.237: -10.406 | <0.001 | 1.005 |
| journal Impact Factor | -1.861 | -2.653: -1.069 | <0.001 | 3.041 |
| Lead lag | R-square: 0.052; Durbin-Watson = 1.620 | | | |
| | Beta | 95%IC | p-value | VIF |
| Study group | -16.966 | -18.570: -15.363 | <0.001 | 1.088 |
| CiteScore (2018) | -5.779 | -6.436: -5.121 | <0.001 | 3.447 |
| Opean Access journal? | -0.097 | -1.930: 1.737 | 0.918 | 1.029 |
| journal Impact Factor | 2.795 | 2.339: 3.251 | <0.001 | 3.550 |
| Indexing lag | R-square: 0.152; Durbin-Watson = 0.550 | | | |
| | Beta | 95%IC | p-value | VIF |
| Study group | 41.721 | 39.374: 44.069 | <0.001 | 1.057 |
| CiteScore (2018) | -0.242 | -0.959: 0.475 | 0.508 | 2.913 |
| Opean Access journal? | 63.494 | 60.803: 66.184 | <0.001 | 1.018 |
| journal Impact Factor | 0.524 | 0.078: 0.969 | 0.021 | 2.968 |

95%IC: 95% confidence interval; VIF: Variance inflation factor.

risks. For instance, the immature publication of a COVID-19 letter to the editor [34] obliged the European Society of Cardiology to immediately release a position statement criticizing that letter to the editor and recommending patients and professionals to maintain their angiotensin receptor blocker treatments. Additionally, the role of these unreviewed preprints in evidence generation through systematic reviews and meta-analyses is not sufficiently elucidated.

In some scientific areas, peer reviewers are paid for their work, which leads an approximately 20% reduction in review time, but the quality of the reviews has not been evaluated [35]. Paying reviewers would increase publishing costs, which would increase subscription rates for authors' institutions or article processing charges for authors. It seems more reasonable to think of strategies to increase invited reviewers' acceptance rate by compensating them in other ways. The publishing system acknowledges three different types of contributors to an article: authors, collaborators, and acknowledged [36]. Although the contribution of a good peer reviewer can have a substantial impact on the final article's version, peer reviewer contribution is only acknowledged with an email and voluntarily recorded in *ad hoc* created registries, which rarely are considered as individual merits. Alternatives to this silent contribution range from including the names of the reviewers in the article's webpage or in the article's first page to including them as part of a collective authorship of an editorial [37]. To increase the visibility and the recognition of a peer reviewer contribution to a given article, bibliographic databases such as PubMed could create a third category of contributorship: the peer reviewer. PubMed currently differentiates authors (retrievable with the [AU] field descriptor) and collaborators (retrievable with the [IR] field descriptor) [38]. Creating a third field descriptor for peer reviewers could increase the recognition of their contribution to the published article.

To be recognized as peer reviewer, the review task should not be performed anonymously. The debate regarding the pros and cons of open peer review is not new [39,40]. However, in recent times, an increasing number of journals have adopted mandatory open peer review, suggesting that anonymous comments can overstate weaknesses and unnecessarily destroy

manuscripts that could be improved with constructive criticism [41]. More importantly, anonymous peer review allows a lack of transparency that favors peer-review fraud. Fake peer review has been associated with both predatory publishers [42] and ethical journals [43]. In our analysis, we found some journals with extremely short acceptance lags: two journals had median acceptance lags lower than 50 days, and 4 journals had 25% of their articles accepted in less than 32 days. It does not seem feasible to select reviewers, deliver their comments, receive the new version of the manuscript from the authors, and evaluate this new version in this short period of time.

Journals have their articles' metadata available in PubMed (including MEDLINE and PubMed Central) only after passing a rigorous selection process by the NLM. One would think that journals accepted in PubMed have the highest quality standards in both their editorial process and in the transparency of this process. However, in our study approximately 73% of pharmacy practice journals and 59% of journals from the comparison group did not include the three dates of their editorial process among the metadata they provide to PubMed. These poor reporting practices were associated with a potential delay underestimation [2]. It is time to ask PubMed and other journal selection committees to mandate that journal publishers provide the dates of their editorial process as a means to ensure publishing transparency.

One should acknowledge that editorial process duration may be different for the different types of publications. However, publication delay analyses usually avoid publication type classification for a number of reasons [8,27,28]. Journals use a non-consistent terminology for publication types. In some journals, short reports include commentaries while other journals include these short reports as letters or research letters. Systematic reviews and meta-analyses are considered as original research papers in some journals and as review articles in other journals. In addition to this inconsistent terminology, publication type indexing is also inconsistent. Journals indexed with metadata provided to Medline can now classify their articles into 30 different publication types. Before February 2016, NLM had only 6 different publication types available to be provided in metadata by the publishers. However, journals indexed with metadata provided to PubMed Central can use their ad hoc created list of publication types. The default value of "Journal Article" will be assigned to any article not providing a valid publication type in their metadata set [44]. Subsequently, limiting bibliometric analyses to articles indexed with specific Publication Types may be misleading. Our analysis tried to avoid this inaccurate selection process and overcome this issue by using a big probabilistic randomly selected sample of articles as comparison group.

Although the "climbing upwards" number of existing journals has been blamed for increasing the number of review invitations [45], a recently published mathematical model demonstrated that the total number of reviewers required to publish one paper is inversely related to the acceptance rate, which should increase in parallel to the number of journals. This inverse relationship means that the higher the number of indexed journals, the lower the rejection rate and subsequently the lower the total number of reviewers necessary to publish a given number of articles [46].

However, educating authors about their important role as peer reviewers is probably the best mid-to-long-term strategy. If authors want to have their manuscripts published in a short time, they must increase their invitation acceptance rate.

## Limitations

Our study has some limitations. The incomplete metadata indexing in PubMed may have influenced the results with unpredictable consequences. We only analyzed the pharmacy practice journals that were identified in Mendes et al.'s study; although they searched the major

databases, they may have not identified all the pharmacy journals. Our study included all the articles published by the journals during the study period, including editorials, which have shorter processing times. However, these contributions should not have an important relative weight in the pool of articles. We only obtained indexing data for PubMed, which may not be generalizable for other bibliographic databases. However, PubMed is the more comprehensive bibliographic database of biomedical literature containing more than 32 million records.

## Conclusion

A pool of articles published in pharmacy practice journals between 2009 and 2018 had similar acceptance lag times to a generic random sample of articles indexed in PubMed: approximately 100 days with a slightly increasing time trend. Small differences in lead time and indexing time do not differentiate the total duration of the editorial process of pharmacy practice journals from that of other journals.

## Supporting information

**S1 Appendix. Comparison group: Results of the PMIDs random selection.**
(DOCX)

**S2 Appendix. Articles published in the pharmacy practice journals between 2009 and 2018.**
(DOCX)

**S3 Appendix. Acceptance, lead, total, and indexing lag times for pharmacy practice journals (2009–2018).**
(DOCX)

**S4 Appendix. Acceptance, lead, total, and indexing lag times trends (2009–2018).**
(DOCX)

**S5 Appendix. Pharmacy practice journals' data (violin plots with lines representing percentile25, median, and percentil75).**
(DOCX)

## Acknowledgments

### Declarations

**Ethics approval.** This study does not require any ethics approval.

## Author Contributions

**Conceptualization:** Fernanda S. Tonin, Roberto Pontarolo, Fernando Fernandez-Llimos.

**Data curation:** Antonio M. Mendes, Felipe F. Mainka, Fernando Fernandez-Llimos.

**Formal analysis:** Antonio M. Mendes, Fernanda S. Tonin, Roberto Pontarolo, Fernando Fernandez-Llimos.

**Methodology:** Fernanda S. Tonin, Fernando Fernandez-Llimos.

**Project administration:** Roberto Pontarolo, Fernando Fernandez-Llimos.

**Supervision:** Roberto Pontarolo, Fernando Fernandez-Llimos.

**Validation:** Antonio M. Mendes.

**Writing – original draft:** Antonio M. Mendes, Fernanda S. Tonin, Roberto Pontarolo, Fernando Fernandez-Llimos.

**Writing – review & editing:** Antonio M. Mendes, Fernanda S. Tonin, Felipe F. Mainka, Roberto Pontarolo, Fernando Fernandez-Llimos.

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
