## [Decision Letter · Decision Letter 0]

16 Mar 2021

PONE-D-21-03317

Publication speed in pharmacy practice journals: a comparative analysis

PLOS ONE

Dear Dr. Fernandez-Llimos,

Thank you for submitting your manuscript to PLOS ONE. After careful consideration, we feel that it has merit but does not fully meet PLOS ONE’s publication criteria as it currently stands. Therefore, we invite you to submit a revised version of the manuscript that addresses the points raised during the review process.

We look forward to receiving your revised manuscript.

Kind regards,

Tim Mathes

Academic Editor

PLOS ONE

Journal Requirements:

2.Thank you for stating the following in the Competing Interests section:

"FFL and FST are editors of the journal Pharmacy Practice. The other authors have no conflict of interest."

Reviewers' comments:

Reviewer's Responses to Questions

**Comments to the Author**

1. Is the manuscript technically sound, and do the data support the conclusions?

Reviewer #1: Partly

Reviewer #2: Partly

2. Has the statistical analysis been performed appropriately and rigorously? 

Reviewer #1: Yes

Reviewer #2: No

3. Have the authors made all data underlying the findings in their manuscript fully available?

Reviewer #1: Yes

Reviewer #2: Yes

4. Is the manuscript presented in an intelligible fashion and written in standard English?

Reviewer #1: Yes

Reviewer #2: Yes

5. Review Comments to the Author

Reviewer #1: The article “Publication speed in pharmacy practice journals: a comparative analysis” by Mendes et al evaluates the duration of the publication process in Pharmacy practice journals compared to other scientific disciplines.

The article provides valuable and significant information with a large sample size of articles and carefully analyzed data. I addition, the article describes some of the pain points of the publication process that add to delays, with peer review as the major process that may significantly add to delays.

The quantification of the acceptance, lead and indexing lags are useful and interesting contributions. Additionally, the comparison of the present results to previous studies such as the Rodriguez data is interesting and sets up a good framework for the need to carry out this particular investigation.

Table 3 with lag times presents useful data to compare the two groups and supports the above notions.

Also interesting is the open access data presented in this study. The magnitude of the indexing lag 92 days vs 5 days for pharmacy (which seems an incredibly large difference…can an 18.4-fold difference be easily explained?)...compared to 4 days vs 3 days in the control seems to ask for further study/data. It would be important to explain this further in the text and perhaps clarify the reasons.

However, the manuscript also has some significant drawbacks. Large sections of text seem opinion based, especially the discussion. The introduction and discussion need to be significantly curtailed and seem to digress significantly from the hard data presented. I would recommend removing large sections of text that are not addressed in the analyses.

During “emergency times”, COVID, the peer review process, pre-prints are all mentioned…all of which are important and currently topical items when exploring publication speed…however, the reader is left with wanting more granular data or at least some direct data that addresses these influencing factors and their effect on publication speed. At best, the current dataset is indirect regarding these discussions. The article should be significantly shortened.

In brief, I would support publication after a very significant reduction in length… perhaps as a short note.

Reviewer #2: Abstract: The phrase “duration of the publication process” seems somewhat ambiguous. This could encompass elements of manuscript preparation all the way to publication of the article itself. Since it seems the rest of the description here focuses on the period between submission date and acceptance date, a better term might relate to the peer review and publication process.

For the comparison group, I think it should be specified whether the comparison group excluded pharmacy practice journals (assuming it did), but should be referred to as “non-pharmacy practice journals” for clarity. Also, it would be helpful to know if these journals potentially included pharmacy journals that did not involve practice (eg, pharmacology, pharmacokinetics, basic science, etc). I think there needs to be some description of the comparison group, eg, broadly speaking the discipline of the journal related to medicine, nursing, public health, basic science, etc. The random selection process seems like it could include any possible article, which would be an extremely heterogeneous group of articles.

Intro, paragraph 2: The discussion of RCT publishing metrics is interesting, but I wonder if there is anything more relevant to pharmacy practice, as pharmacists typically are not performing RCTs, but rather observational or pre-post studies.

Intro paragraph 3: The critique of the peer review process in the context of this research might be strengthened by statements that relate more closely to the duration of this process, since that seems to be the main outcome measure of this study.

Intro, paragraph 4: The final sentence of this paragraph relates to one of my earlier comments regarding the composition of the comparator group. Here, the authors state they aim to compare pharmacy practice journals with “other biomedical disciplines”. The other biomedical disciplines that composed the comparator group should be described in the abstract.

Data collection, paragraph 2: The authors aim to identify a 10-day difference between groups in publication time. I think this selection of a clinically important difference requires justification. This seems like a very short period and is well below the SD and IQR of analyses of pharmacy practice journals themselves. I think this may be too short of a period, considering the need for NLM to distribute resources responsibly for indexing, and the lower likelihood that pharmacy practice journals will publish something practice-changing as compared to general medical journals. If anything, this will only leave the analysis with greater power rather than less, but the selection requires justification. I also think this paragraph needs disambiguation of the term “article reception”.

Data process, paragraph 1: I think the terms related to the Medline metadata require definition to assist the readers in knowing how they differ. For example, it may be easy for a reader to not appreciate the difference between acceptance date and entry date. Additionally, I think some more context is needed regarding the terminology of “indexing lag”. This term might be interpreted as “indexing” with MeSH headings (ie, MHDA data in MEDLINE), but it seems that the authors are using Entrez date (EDAT data in MEDLINE), which is generally speaking the online publication date.

Data analysis: I’m not entirely certain of the benefit of presenting a Cohen’s d in this case. It seems to complicate the interpretation somewhat as its calculation loses the units of the measure. I think it would be more beneficial to report a point estimate and confidence interval for the difference between groups. After having read the remainder of the results, the authors (appropriately) have evaluated the effect of other variables on the outcome of various time periods. With this in mind, I wonder if a multivariable model adjusting for the variables while predicting each outcome would be beneficial. Alternatively, I was almost anticipating some matching procedure between pharmacy practice journals and comparator journals to ensure they have some similar characteristics (eg, MeSH headings for comparative study, meta-analysis, etc) to try to answer if there is something related to the journal discipline that delays this process, all other things being equal. Lastly, since time is an element to this analysis, I wonder if a survival analysis is more appropriate. If there are any exclusions for articles that do not have one of the “outcome” variables, it presents risk for bias in the analysis of simply comparing median times.

Results, paragraph 2: The authors describe the languages of journals in the comparison group here, but do not do so for the pharmacy practice “exposure” group. I think this should be reported, as any differences between the groups in language could influence availability of peer reviewers or other processes in the duration of the process the authors are measuring.

Table 1: I’m not sure how beneficial columns 2 through 4 are in this table. I think the most helpful information is knowing the distribution of publication dates for each group over time. Eg, I do not think there is much that readers can take away from knowing the first PMID that was eligible for inclusion from each 1-year period.

Table 2: The content in this table has confused me a bit regarding the methodology. I see that not all articles in either group consistently reported important data for the three variables. So have the articles without this data been excluded from analyses? I don’t recall reading that in the methods, so if this is the case, I believe it should be specified, or otherwise clarified.

Paragraph following table 2: The information on time trend is interesting, but I think needs more interpretation for readers. The correlation coefficient seems difficult to interpret in this situation – eg, it may help to explain the average annual improvement in lag time for each group.

Table 3: Similar to my above comment, I think instead of reporting Cohen’s d, a point estimate and confidence interval for a difference would be more interpretable and helpful.

Paragraph following table 3: These trends within pharmacy journals are interesting, but I think their presentation needs more justification. In other words, why mention these specific journals? Did they have the shortest and longest acceptance lags, or were in the upper and lower deciles? Some type of criterion for presentation should be specified. It would also help to include results for each journal in tabular format (though in a supplemental table, I imagine), potentially ordered by impact factor or some other metric of readership. On this topic, I find it a shortcoming that some of the more widely read pharmacy practice journals (eg, AJHP, Pharmacotherapy) do not have their results reported to serve as a barometer for pharmacists who are likely most familiar with these.

Discussion: This section is rather long given the scope of the article. The first three paragraphs are relevant in providing additional context to the findings of this analysis. The remainder, while offering some interesting insight and potential solutions, can be tapered somewhat. I also get the impression here more than in other sections of the manuscript that the authors take most issue with delays during the peer review process. Based on the definitions of each lag time, it seems that acceptance lag may be most representative of the peer review process. The authors might consider restricting the analysis and presentation to this measure only to give the article a tighter focus.

Limitations: I think a major limitation that should be stated is the heterogeneous nature of the comparison group. While I think the comparison group should be modified with matching or other techniques as described above, I think it is difficult to compare essentially pharmacy practice journals with “everything else”. There are so many journals in Pubmed that are not directly related to healthcare provision (eg, basic science) that it may make this comparison group somewhat unrepresentative.

6. PLOS authors have the option to publish the peer review history of their article (what does this mean?). If published, this will include your full peer review and any attached files.

Reviewer #1: No

Reviewer #2: No

---

## [Author Response · Author response to Decision Letter 0]

16 Apr 2021

Reviewer #1: 

The article “Publication speed in pharmacy practice journals: a comparative analysis” by Mendes et al evaluates the duration of the publication process in Pharmacy practice journals compared to other scientific disciplines.

The article provides valuable and significant information with a large sample size of articles and carefully analyzed data. I addition, the article describes some of the pain points of the publication process that add to delays, with peer review as the major process that may significantly add to delays.

The quantification of the acceptance, lead and indexing lags are useful and interesting contributions. Additionally, the comparison of the present results to previous studies such as the Rodriguez data is interesting and sets up a good framework for the need to carry out this particular investigation.

Table 3 with lag times presents useful data to compare the two groups and supports the above notions.

Also interesting is the open access data presented in this study. The magnitude of the indexing lag 92 days vs 5 days for pharmacy (which seems an incredibly large difference…can an 18.4-fold difference be easily explained?)...compared to 4 days vs 3 days in the control seems to ask for further study/data. It would be important to explain this further in the text and perhaps clarify the reasons.

However, the manuscript also has some significant drawbacks. Large sections of text seem opinion based, especially the discussion. The introduction and discussion need to be significantly curtailed and seem to digress significantly from the hard data presented. I would recommend removing large sections of text that are not addressed in the analyses.

During “emergency times”, COVID, the peer review process, pre-prints are all mentioned…all of which are important and currently topical items when exploring publication speed…however, the reader is left with wanting more granular data or at least some direct data that addresses these influencing factors and their effect on publication speed. At best, the current dataset is indirect regarding these discussions. The article should be significantly shortened.

In brief, I would support publication after a very significant reduction in length… perhaps as a short note.

RE: Thank you very much for your kind and encouraging comments. We could not reduce the length of the paper because reviewer #2 asked for additional information in several comments.

 

Reviewer #2: 

Abstract: The phrase “duration of the publication process” seems somewhat ambiguous. This could encompass elements of manuscript preparation all the way to publication of the article itself. Since it seems the rest of the description here focuses on the period between submission date and acceptance date, a better term might relate to the peer review and publication process.

RE: In fact, we analyzed all the entire publication process, from the submission to the indexing. In the sub-section ‘Data process’, we explained the four different lag times we calculated: total publication lag, acceptance lag, lead lag, and indexing lag. The time between the submission date and acceptance date is covered by the acceptance lag, being the remaining lag times other parts of the editorial process. Figure1 depicts these time intervals.

For the comparison group, I think it should be specified whether the comparison group excluded pharmacy practice journals (assuming it did), but should be referred to as “non-pharmacy practice journals” for clarity. Also, it would be helpful to know if these journals potentially included pharmacy journals that did not involve practice (eg, pharmacology, pharmacokinetics, basic science, etc). I think there needs to be some description of the comparison group, eg, broadly speaking the discipline of the journal related to medicine, nursing, public health, basic science, etc. The random selection process seems like it could include any possible article, which would be an extremely heterogeneous group of articles.

RE: Thank you very much for this comment. We double-checked all the 23888 articles that we had originally included in the comparison group, and realized that a few were pharmacy practice journal articles. As they were a reduced group, results were virtually not affected by the deletion of these articles from the comparison group, but we have re-done all the calculations. So, now we can ensure that both groups are mutually exclusive. We clarified this in Methods section. Regarding the potential heterogeneity in this non-pharmacy practice group, we have intentionally aimed to compare pharmacy practice journals with all the non-pharmacy practice biomedical journals. In a previous research (doi: 10.1016/j.sapharm.2016.01.003), we demonstrated that pharmacy is not an homogeneous scientific area, and we cannot infer that non-practice pharmacy journals perform closer to pharmacy practice journals than to other biomedical journals. Probably, non-practice pharmacy journals may not be a category itself different to other biomedical areas (i.e., clinical, molecular or cell pharmacology are areas shared by pharmacists and physicians).

Intro, paragraph 2: The discussion of RCT publishing metrics is interesting, but I wonder if there is anything more relevant to pharmacy practice, as pharmacists typically are not performing RCTs, but rather observational or pre-post studies.

RE: As far as we know, this is the first study evaluating editorial process duration in pharmacy practice. So, we could not find any published data to frame the scene other than clinical trials. In paragraph 4 of the introduction we mentioned the results of Rodriguez assessing one of the steps of the editorial process we evaluated as a whole. Additionally, we cannot concur with the final statement of this comment, as per our previous research (doi: 10.1016/j.sapharm.2016.01.003). 

Intro paragraph 3: The critique of the peer review process in the context of this research might be strengthened by statements that relate more closely to the duration of this process, since that seems to be the main outcome measure of this study.

RE: In intro paragraph 3 we have not criticized the peer review process. We are strongly supportive of peer review as a basic requirement for robust and reliable scholarly publishing. That’s why we quoted the Kassirer sentence “crude and understudied, but indispensable”[10]. We devoted paragraphs 4, 5, and 6 of the discussion to comment potential improvements to the peer review process.

Intro, paragraph 4: The final sentence of this paragraph relates to one of my earlier comments regarding the composition of the comparator group. Here, the authors state they aim to compare pharmacy practice journals with “other biomedical disciplines”. The other biomedical disciplines that composed the comparator group should be described in the abstract.

RE: We reworded the sentence to clarify this in the abstract.

Data collection, paragraph 2: The authors aim to identify a 10-day difference between groups in publication time. I think this selection of a clinically important difference requires justification. This seems like a very short period and is well below the SD and IQR of analyses of pharmacy practice journals themselves. I think this may be too short of a period, considering the need for NLM to distribute resources responsibly for indexing, and the lower likelihood that pharmacy practice journals will publish something practice-changing as compared to general medical journals. If anything, this will only leave the analysis with greater power rather than less, but the selection requires justification. I also think this paragraph needs disambiguation of the term “article reception”.

RE: We added a sentence in the mentioned paragraph. We performed a preliminary analysis and found, as mentioned in the sample size paragraph, a “mean delay of 125 days (SD 98) between article reception and publication”, which represents the entire publication process lag. 10 days represent the 8% of that time, which we considered appropriate to consider as difference to establish sample size. We agree with the reviewer that short period we selected have increased the statistical power or our analyses. But we specifically needed that power to ascertain differences between the two groups lead lag (13 vs. 23 days) or the indexing lag (5 vs. 4 days). But we again cannot concur with the reviewer consideration of pharmacy practice field, after reviewer’s statement “lower likelihood that pharmacy practice journals will publish something practice-changing as compared to general medical journals”. Pharmacy practice journals will surely not publish papers medical practice-changing, but will publish more pharmacy practice-changing papers than medical journals.

Data process, paragraph 1: I think the terms related to the Medline metadata require definition to assist the readers in knowing how they differ. For example, it may be easy for a reader to not appreciate the difference between acceptance date and entry date. Additionally, I think some more context is needed regarding the terminology of “indexing lag”. This term might be interpreted as “indexing” with MeSH headings (ie, MHDA data in MEDLINE), but it seems that the authors are using Entrez date (EDAT data in MEDLINE), which is generally speaking the online publication date.

RE: We added a short description for all the metadata retrieved from PubMed. We also added a sentence to clarify why we have not calculated the Cataloging lag (time to allocate MeSH). 

Data analysis: I’m not entirely certain of the benefit of presenting a Cohen’s d in this case. It seems to complicate the interpretation somewhat as its calculation loses the units of the measure. I think it would be more beneficial to report a point estimate and confidence interval for the difference between groups. After having read the remainder of the results, the authors (appropriately) have evaluated the effect of other variables on the outcome of various time periods. With this in mind, I wonder if a multivariable model adjusting for the variables while predicting each outcome would be beneficial. Alternatively, I was almost anticipating some matching procedure between pharmacy practice journals and comparator journals to ensure they have some similar characteristics (eg, MeSH headings for comparative study, meta-analysis, etc) to try to answer if there is something related to the journal discipline that delays this process, all other things being equal. Lastly, since time is an element to this analysis, I wonder if a survival analysis is more appropriate. If there are any exclusions for articles that do not have one of the “outcome” variables, it presents risk for bias in the analysis of simply comparing median times.

RE: We used Cohen’s d following ASA recommendations, instead of Null Hypothesis tests (mentioned in paragraph 4 of the results). We completely agree with the reviewer that many additional calculations can be made to find out potential predictors of publication delay. In fact, we aim to continue this analysis with publication country as predictor, which in our preliminary analyses showed promising results. In this piece of research, we aimed to make a descriptive analysis of pharmacy practice versus the other biomedical areas. 

Results, paragraph 2: The authors describe the languages of journals in the comparison group here, but do not do so for the pharmacy practice “exposure” group. I think this should be reported, as any differences between the groups in language could influence availability of peer reviewers or other processes in the duration of the process the authors are measuring.

RE: Thank you. This was an unintentional omission that we have now corrected. Having both groups with very similar English percentage, language may not be a major influence on the acceptance lag difference.

Table 1: I’m not sure how beneficial columns 2 through 4 are in this table. I think the most helpful information is knowing the distribution of publication dates for each group over time. Eg, I do not think there is much that readers can take away from knowing the first PMID that was eligible for inclusion from each 1-year period.

RE: We agree that Table 1 provided excessive information for readers, so we deleted columns 2 to 4.

Table 2: The content in this table has confused me a bit regarding the methodology. I see that not all articles in either group consistently reported important data for the three variables. So have the articles without this data been excluded from analyses? I don’t recall reading that in the methods, so if this is the case, I believe it should be specified, or otherwise clarified.

RE: We made the sample size calculations considering the poor metadata reporting we found in the preliminary analyses (and we mentioned “with 43.4% of the articles providing data for this calculation”). We have now added a sentence in Data Analysis section to further clarify.

Paragraph following table 2: The information on time trend is interesting, but I think needs more interpretation for readers. The correlation coefficient seems difficult to interpret in this situation – eg, it may help to explain the average annual improvement in lag time for each group.

RE: We agree with the reviewer that correlation coefficients may not be easily understood by some readers. However, as the trends were not constant, which may difficult the calculation of average improvements, we preferred adding a new online appendix (Supporting information 4) providing medians and IQR for each year.

Table 3: Similar to my above comment, I think instead of reporting Cohen’s d, a point estimate and confidence interval for a difference would be more interpretable and helpful.

RE: We preferred following American Statistical Association recommendations provided in 2019 supplementary issue, and specially Wasserstein & Lazar “Moving to a World Beyond “p < 0.05””, and provide an effects size measure to show the size of the difference between the two distributions.

Paragraph following table 3: These trends within pharmacy journals are interesting, but I think their presentation needs more justification. In other words, why mention these specific journals? Did they have the shortest and longest acceptance lags, or were in the upper and lower deciles? Some type of criterion for presentation should be specified. It would also help to include results for each journal in tabular format (though in a supplemental table, I imagine), potentially ordered by impact factor or some other metric of readership. 

RE: We agree we had not explained why we mentioned those journals. They are the extremes of the distributions. We have now reworded the sentence to clarify. The results for each journal are provided in Supporting information – S 5.

On this topic, I find it a shortcoming that some of the more widely read pharmacy practice journals (eg, AJHP, Pharmacotherapy) do not have their results reported to serve as a barometer for pharmacists who are likely most familiar with these.

RE: American journal of Hospital Pharmacy, now called American Journal of Health-System Pharmacy, is included in the analyses. Unfortunately, this journal is one of those not reporting editorial process dates in their metadata submitted to PubMed. Regarding Pharmacotherapy, we have used an objective method to select journals to analyze (Res Social Adm Pharm. 2019;15:1464-71.). Pharmacotherapy appears as clinical pharmacology journals, and not pharmacy practice journal, so it is not included in our analysis.

Discussion: 

This section is rather long given the scope of the article. The first three paragraphs are relevant in providing additional context to the findings of this analysis. The remainder, while offering some interesting insight and potential solutions, can be tapered somewhat. I also get the impression here more than in other sections of the manuscript that the authors take most issue with delays during the peer review process. Based on the definitions of each lag time, it seems that acceptance lag may be most representative of the peer review process. The authors might consider restricting the analysis and presentation to this measure only to give the article a tighter focus.

RE: We aimed to evaluate all the lag times in the editorial process. We agree that acceptance lag is the greatest delay in the process (as demonstrated by our results). Consequently, our discussion focused on the main cause of the duration of this lag time, the peer review process. Not by chance, peer-review process is having major attention in literature, and many alternatives (some cited in our discussion) are been suggested to overcome this issue. We’d appreciate the revierwer could be more specific guiding us in what should we reduce in the discussion.

Limitations: I think a major limitation that should be stated is the heterogeneous nature of the comparison group. While I think the comparison group should be modified with matching or other techniques as described above, I think it is difficult to compare essentially pharmacy practice journals with “everything else”. There are so many journals in Pubmed that are not directly related to healthcare provision (eg, basic science) that it may make this comparison group somewhat unrepresentative.

RE: We humbly disagree with the reviewer. Our aim was to compare the editorial process in pharmacy journals with what happens in other biomedical fields, not with specific fields. We added a sentence in Methods to explain why we selected PubMed as the source of the comparison group. We also added a justification to the already existing limitation about the use of a random sample of PubMed articles.

---

## [Decision Letter · Decision Letter 1]

4 May 2021

PONE-D-21-03317R1

Publication speed in pharmacy practice journals: a comparative analysis

PLOS ONE

Dear Dr. Fernandez-Llimos,

Thank you for submitting your manuscript to PLOS ONE. After careful consideration, we feel that it has merit but does not fully meet PLOS ONE’s publication criteria as it currently stands. Therefore, we invite you to submit a revised version of the manuscript that addresses the points raised during the review process.

 A premise to be considered for publication is a revision of the statistical analyses (compare comment of reviewer 2 regarding the statistical analyses). To facilitate interpretation it would be better to perform the analysis on the original scale, instead of using  Cohen’s d. Furthermore, a multivariate analysis should be performed because it can be expected that the univariate analyses are confounded and consequently it is not possible to assess the impact of individual factors.

We look forward to receiving your revised manuscript.

Kind regards,

Tim Mathes

Academic Editor

PLOS ONE

Reviewers' comments:

Reviewer's Responses to Questions

**Comments to the Author**

1. If the authors have adequately addressed your comments raised in a previous round of review and you feel that this manuscript is now acceptable for publication, you may indicate that here to bypass the “Comments to the Author” section, enter your conflict of interest statement in the “Confidential to Editor” section, and submit your "Accept" recommendation.

Reviewer #1: All comments have been addressed

2. Is the manuscript technically sound, and do the data support the conclusions?

Reviewer #1: Yes

3. Has the statistical analysis been performed appropriately and rigorously? 

Reviewer #1: Yes

4. Have the authors made all data underlying the findings in their manuscript fully available?

Reviewer #1: Yes

5. Is the manuscript presented in an intelligible fashion and written in standard English?

Reviewer #1: Yes

6. Review Comments to the Author

Reviewer #1: The authors have addresses the queries I had made. Although they disagree with some of my comments, they have provided an explanation and have made some minor modifications that have improved the paper.

7. PLOS authors have the option to publish the peer review history of their article (what does this mean?). If published, this will include your full peer review and any attached files.

Reviewer #1: No

---

## [Author Response · Author response to Decision Letter 1]

5 Jun 2021

Response to the editor:

Thank you for submitting your manuscript to PLOS ONE. After careful consideration, we feel that it has merit but does not fully meet PLOS ONE’s publication criteria as it currently stands. Therefore, we invite you to submit a revised version of the manuscript that addresses the points raised during the review process.

A premise to be considered for publication is a revision of the statistical analyses (compare comment of reviewer 2 regarding the statistical analyses). To facilitate interpretation it would be better to perform the analysis on the original scale, instead of using Cohen’s d. Furthermore, a multivariate analysis should be performed because it can be expected that the univariate analyses are confounded and consequently it is not possible to assess the impact of individual factors.

RE: Thank you very much for the opportunity to improve our manuscript. Following previous recommendations from reviewer #2 we have now included the multivariate analyses by performing a multivariate linear regression for the three lag times. The results, presented in Table 4, confirm the findings reported with bivariate analysis.

Table 4. Multivariate analyses for the three lag times 

Acceptance lag R-square: 0.006; Durbin-Watson= 1.788

 Beta 95%IC p-value VIF

Study group -3.910 -7.077 : -0.742 0.016 1.072

CiteScore (2018) 1.000 -0.267 : 2.268 0.122 2.991

Opean Access journal? -13.822 -17.237 : -10.406 <0.001 1.005

journal Impact Factor -1.861 -2.653 : -1.069 <0.001 3.041

Lead lag R-square: 0.052; Durbin-Watson= 1.620

 Beta 95%IC p-value VIF

Study group -16.966 -18.570 : -15.363 <0.001 1.088

CiteScore (2018) -5.779 -6.436 : -5.121 <0.001 3.447

Opean Access journal? -0.097 -1.930 : 1.737 0.918 1.029

journal Impact Factor 2.795 2.339 : 3.251 <0.001 3.550

Indexing lag R-square: 0.152; Durbin-Watson= 0.550

 Beta 95%IC p-value VIF

Study group 41.721 39.374 : 44.069 <0.001 1.057

CiteScore (2018) -0.242 -0.959 : 0.475 0.508 2.913

Opean Access journal? 63.494 60.803 : 66.184 <0.001 1.018

journal Impact Factor 0.524 0.078 : 0.969 0.021 2.968

95%IC: 95% confidence interval; VIF: Variance inflation factor

Regarding the use of Cohen’d d in table 2, the comment Reviewer #2 provided in the previous round stated: “Table 3: Similar to my above comment, I think instead of reporting Cohen’s d, a point estimate and confidence interval for a difference would be more interpretable and helpful”. To attend this comment, we included the median (as a point estimate) and the IQR (instead the confidence interval, which we cannot use because we are presenting dispersion measure of a non-normally distributed variable). Additionally, we presented the null hypothesis test that that estimated the significance of the differences between the two groups under analysis. These two elements comply with the requirements of the reviewer #2. In addition to these requested elements, and following the recommendations of the American Statistical Association, we insist in providing the effect size measures of these differences, presented as the Cohen’s d. The latter should be understood as additional information after providing the requested information.

---

## [Editor Report · Decision Letter 2]

11 Jun 2021

Publication speed in pharmacy practice journals: a comparative analysis

PONE-D-21-03317R2

Dear Dr. Fernandez-Llimos,

We’re pleased to inform you that your manuscript has been judged scientifically suitable for publication and will be formally accepted for publication once it meets all outstanding technical requirements.

Kind regards,

Tim Mathes

Academic Editor

PLOS ONE
---

## [Editor Report · Acceptance letter]

17 Jun 2021

PONE-D-21-03317R2 

Publication speed in pharmacy practice journals: a comparative analysis 

Dear Dr. Fernandez-Llimos:

I'm pleased to inform you that your manuscript has been deemed suitable for publication in PLOS ONE. Congratulations! Your manuscript is now with our production department. 

Kind regards, 

on behalf of

Dr. Tim Mathes 

Academic Editor

PLOS ONE